# Nanofat Accelerates and Improves the Vascularization, Lymphatic Drainage and Healing of Full-Thickness Murine Skin Wounds

**DOI:** 10.3390/ijms25020851

**Published:** 2024-01-10

**Authors:** Ettore Limido, Andrea Weinzierl, Emmanuel Ampofo, Yves Harder, Michael D. Menger, Matthias W. Laschke

**Affiliations:** 1Institute for Clinical and Experimental Surgery, Saarland University, 66421 Homburg, Germany; limidoettore@gmail.com (E.L.); andreaweinzierl@icloud.com (A.W.); emmanuel.ampofo@uks.eu (E.A.); michael.menger@uks.eu (M.D.M.); 2Department of Plastic Surgery and Hand Surgery, University Hospital Zurich, 8091 Zurich, Switzerland; 3Department of Plastic, Reconstructive and Aesthetic Surgery, Ospedale Regionale di Lugano, Ente Ospedaliero Cantonale (EOC), 6900 Lugano, Switzerland; yves.harder@yahoo.com; 4Faculty of Biomedical Sciences, Università della Svizzera Italiana, 6900 Lugano, Switzerland

**Keywords:** wound healing, nanofat, platelet-rich plasma, vascularization, angiogenesis, lymph vessels, macrophages

## Abstract

The treatment of wounds using the body’s own resources is a promising approach to support the physiological regenerative process. To advance this concept, we evaluated the effect of nanofat (NF) on wound healing. For this purpose, full-thickness skin defects were created in dorsal skinfold chambers of wild-type mice. These defects were filled with NF generated from the inguinal subcutaneous adipose tissue of green fluorescent protein (GFP)^+^ donor mice, which was stabilized using platelet-rich plasma (PRP). Empty wounds and wounds solely filled with PRP served as controls. Wound closure, vascularization and formation of granulation tissue were repeatedly analyzed using stereomicroscopy, intravital fluorescence microscopy, histology and immunohistochemistry over an observation period of 14 days. PRP + NF-treated wounds exhibited accelerated vascularization and wound closure when compared to controls. This was primarily due to the fact that the grafted NF contained a substantial fraction of viable GFP^+^ vascular and lymph vessel fragments, which interconnected with the GFP^−^ vessels of the host tissue. Moreover, the switch from inflammatory M1- to regenerative M2-polarized macrophages was promoted in PRP + NF-treated wounds. These findings indicate that NF markedly accelerates and improves the wound healing process and, thus, represents a promising autologous product for future wound management.

## 1. Introduction

Wound healing is a dynamic biological process that aims to restore the anatomical and functional integrity of the skin through the tightly regulated interaction of several cell types and mediators. This process is characterized by three distinct but overlapping phases [1]. During the first hemostatic/inflammatory phase, the activation of the hemostatic cascade and platelets leads to clot formation and a massive release of chemokines. This, in turn, increases the vascular permeability and promotes the recruitment of neutrophils and macrophages to the wound site. Macrophages in particular play a major role in the transition from the inflammatory to the secondary proliferative phase, in which the hemostatic clot and tissue debris are gradually replaced by highly vascularized granulation tissue [2]. The third remodeling phase is then characterized by the regression of microvessels and the formation of a collagen-rich scar tissue [3].

An Impaired wound healing process may result in the development of chronic wounds, especially in the presence of associated comorbidities, such as diabetes, infections or chronic venous insufficiency [4,5]. For the treatment of such wounds, several therapeutic strategies have already been established in clinical practice, including advanced wound dressings, negative wound pressure therapy and hyperbaric oxygen wound therapy [6]. In addition, preclinical studies showed the potential of growth factor release systems or isolated pluripotent stem cells to improve wound healing [7,8]. However, the latter biological approaches have the disadvantages that they are time-consuming, are cost-intensive and underly strict ethical and regulatory hurdles [9].

These problems may be overcome by the use of nanofat (NF), which is applied in the field of regenerative and aesthetic medicine for facial rejuvenation, lipomodeling, scar repair, burn wound management and alopecia treatment [10,11,12,13]. NF is an autologous fat derivative that is generated by mechanical emulsification and filtration to obtain a liquid suspension [14]. In this suspension, most of the mature adipocytes are destroyed [15,16]. Accordingly, it is mainly characterized by a high content of extracellular matrix, growth factors, adipose tissue-derived stem cells (ADSCs) and microvascular fragments (MVFs) [17,18,19]. All of these individual components have already been shown to promote tissue formation and the development of new blood-perfused microvascular networks [20,21,22]. Therefore, it may be assumed that NF ideally supports the regeneration of skin defects.

To test this hypothesis, we analyzed in the present proof-of-concept study the effects of NF on wound healing. For this purpose, we mixed NF from donor mice with platelet-rich plasma (PRP) to fix it in full-thickness skin wounds that were created in mouse dorsal skinfold chambers. The healing of these wounds was analyzed by means of stereomicroscopy, intravital fluorescence microscopy as well as histological and immunohistochemical techniques in comparison to empty wounds and wounds that were solely treated with PRP.

## 2. Results

### 2.1. Generation and Characterization of NF

We generated NF from fat samples of green fluorescent protein (GFP)^+^ donor mice (Figure 1A). For this purpose, their inguinal subcutaneous adipose tissue was excised and mechanically processed by a tissue chopper and a female-to-female Luer lock system (Figure 1B,C). Immunohistochemical analyses of this NF showed a mixture of perilipin^+^ disrupted and intact adipocytes as well as collagen (Col) I^+^ and III^+^ extracellular matrix components (Figure 1D–F). Importantly, NF also contained fragments of both CD31^+^ microvessels and lymphatic vessel endothelial hyaluronan receptor (LYVE)-1^+^ lymphatic vessels (Figure 1G,H).

### 2.2. Repeated In Vivo Microscopy of Healing Wounds

The effects of NF on wound healing were repeatedly analyzed in full-thickness skin defects, which were created in the dorsal skinfold chamber of GFP^−^ recipient mice (Figure 1I–L). Stereomicroscopic imaging showed the progressive healing of empty, PRP- and PRP + NF-treated wounds throughout the observation period of 14 days (Figure 2A). Of interest, wound healing was accelerated in the PRP + NF group, as indicated by a reduced wound area on day 10 and 14 when compared to the other two groups (Figure 2A,B).

In addition, the vascularization of the wounds was investigated by means of intravital fluorescence microscopy (Figure 3). Of note, the pre-existing blood vessels in the wound beds could be clearly visualized in empty and PRP-treated wounds on day 0, whereas this was not the case in PRP + NF-treated wounds due to the opacity of the incorporated GFP^+^ NF (Figure 3A). This, however, did not affect the subsequent analysis of ingrowing new blood vessels from the wound borders. The analysis showed that only PRP + NF-treated wounds contained newly formed microvessels on day 3 (Figure 3B). Accordingly, they also exhibited a higher number of perfused regions of interest (ROIs) on day 3 and 6 when compared to empty wounds (Figure 3B). On day 10, wounds filled with PRP and PRP + NF presented with a higher number of perfused ROIs when compared to empty controls (Figure 3B). This indicates that PRP itself also slightly promoted the vascularization of the wounds due to its intrinsic angiogenic activity. More detailed analyses revealed a significantly higher functional microvessel density in PRP + NF-treated wounds between day 3 and 10 (Figure 3C), whereas microhemodynamic parameters did not markedly differ between the three experimental groups throughout the 14-day observation period (Table 1). Taken together, these findings demonstrate that the combination of PRP and NF most effectively accelerates the vascularization and closure of wounds.

### 2.3. Histological and Immunohistochemical Analyses of Wounds

At the end of the in vivo experiments on day 14, the healing wounds were additionally analyzed by means of histology and immunohistochemistry. In line with our in vivo results, PRP + NF-treated wounds finally exhibited increased epithelialization, granulation tissue formation and cellular density when compared to empty wounds (Figure 4A–D). There were no significant differences in the Col I ratio between the experimental groups (Figure 4E,F). However, PRP + NF-treated wounds displayed a markedly higher amount of Col III when compared to the other two groups (Figure 4E,G).

Furthermore, PRP + NF-treated wounds contained a significantly higher number of CD31^+^ microvessels when compared to empty wounds (Figure 5A,B). Of interest, ~30% of these microvessels were GFP^+^, indicating their origin from the GFP^+^ microvessel fragments inside the grafted NF (Figure 5C,D). In addition, PRP + NF-treated wounds also exhibited a higher density of LYVE-1^+^ lymph vessels, and ~20% of these lymph vessels were GFP^+^ (Figure 5E–H). 

Finally, to evaluate the number of immune cells infiltrating the healing wounds, the fraction of M1- and M2-polarized macrophages was assessed by means of CD86 and CD163 stainings (Figure 6A–D). PRP + NF-treated wounds contained the lowest fraction of CD86^+^ M1 macrophages and the highest fraction of CD163^+^ M2 macrophages (Figure 6B,C). Accordingly, they also exhibited an increased CD163^+^/CD86^+^ ratio when compared to empty and PRP-treated wounds (Figure 6D).

## 3. Discussion

NF was initially introduced into clinical practice by Tonnard et al. [14] as a mechanically processed fat derivative, which is now broadly used in aesthetic medicine for the treatment of wrinkles and scars [23,24,25]. This is partially due to its high content of ADSCs, promoting tissue regeneration through both angiogenic growth factor secretion and multi-lineage cell differentiation [18]. In addition, NF contains functional MVFs, which have already been shown to boost the vascularization of implanted biomaterials and tissue substitutes in different experimental settings [17,26]. Hence, NF represents a promising biological product to promote the healing of wounds by means of the body’s own resources.

Chen et al. [27] already reported beneficial effects of NF on the healing of wounds in diabetic rats. For this purpose, they injected small volumes of the fat derivative at four different sites into the wound margins. While this approach may be suitable for the treatment of small skin defects, larger wounds may require another method of NF application that allows the full coverage of the entire wound bed. In this case, it is necessary to stabilize the NF due to its liquid consistency. To achieve this, we herein mixed NF with PRP. This autologous blood derivative bears the advantage that it forms a stable clot after thrombin activation and contains a mixture of various angiogenic growth factors, including transforming growth factor-β, fibroblast growth factor-2, platelet-derived growth factors, insulin-like growth factor-1, epidermal growth factor and hepatocyte growth factor [28,29,30]. Accordingly, we also detected a slightly improved vascularization of PRP-treated wounds when compared to empty controls. However, this beneficial effect on the wound healing process was less pronounced when compared to the PRP + NF group. In fact, in the PRP + NF group, we detected a significantly higher number of perfused ROIs on day 3 as well as an increased functional microvessel density between day 3 and 10 and centerline RBC velocity and shear rate on day 6 when compared to the PRP group. This may be explained by the fact that in addition to angiogenic growth factors deriving from PRP, NF also contains many MVFs [19]. These functional vessel segments exhibit the unique ability to rapidly reassemble into new microvascular networks and to develop functional interconnections with the surrounding host microvasculature [31]. In line with this finding, our intravital fluorescence microscopic analyses showed blood-perfused microvessels in PRP + NF-treated wounds already on day 3, whereas empty and PRP-treated wounds still lacked any blood perfusion at this early time point after setting the skin defect. In addition, immunohistochemical analyses of PRP + NF-treated wounds revealed that they finally contained ~30% of CD31^+^/GFP^+^ microvessels, which originated from MVFs within the GFP^+^ NF from the donor mice.

Moreover, we detected ~20% of LYVE-1^+^/GFP^+^ lymph vessels in PRP + NF wounds. This is an interesting result, since it demonstrates for the first time that NF is also a source for lymph vessels. In our experimental setting, they obviously survived the post-transplantation phase and actively contributed to a higher lymph vessel density inside the treated wounds. Considering the fact that lymph vessel formation is an integral part of wound healing [17,32,33], this mechanism may have further contributed to the enhanced epithelialization, formation of granulation tissue and cellular density of PRP + NF-treated wounds. Beyond this, our novel finding suggests that the use of NF may also be an attractive approach for the treatment of conditions that are associated with a congenital or acquired dysfunctional lymphatic system and lymph edema.

Another major advantage of filling the wound bed with NF is the fact that it also contains Col I and Col III, as shown by our immunohistochemical analyses. These extracellular matrix components may serve as a natural scaffold for the invasion of cells and blood vessels into the wound bed and, thus, may further promote the formation of granulation tissue. The initial phase of the wound healing process is characterized by increased levels of Col III [34]. We detected significantly higher amounts of Col III in PRP + NF-treated wounds on day 14 when compared to PRP-treated and empty wounds. Therefore, we assume that at this time point, even the PRP + NF-treated wounds were still in an active, proliferative phase and had not yet entered the remodeling stage, which is typically associated with higher Col I levels [35].

Finally, we investigated the tissue infiltration by macrophages. By now, it is known that M1-polarized macrophages are predominantly found in the inflammatory phase of the wound healing process [36]. In contrast, M2-polarized macrophages are particularly present during the proliferative phase [2]. Of interest, we detected a higher number of M1-polarized macrophages and a lower number M2-polarized macrophages in empty wounds when compared to PRP-treated and PRP + NF-treated wounds. This indicates that macrophage polarization within the wounds was mainly mediated by PRP. In line with this view, Uchiyama et al. [37] recently demonstrated that PRP promotes M2 polarization. However, the switch from M1 to M2 macrophages was even more pronounced in PRP + NF-treated wounds when compared to PRP-treated wounds. This can be explained by a study of Heo et al. [38] showing that ADSCs, which represent a major cell population in NF, induce the polarization from M1 to M2 macrophages.

Taken together, this proof-of-concept study demonstrates for the first time that NF can be stabilized by means of PRP in the wound bed, where it markedly accelerates and improves tissue regeneration. The fact that NF and PRP are already well-established in clinical practice and both biological products can be harvested by minimally invasive procedures make this strategy attractive for future wound management.

## 4. Materials and Methods

### 4.1. Animals

For the generation of PRP, adult male and female C57BL/6J mice (Institute for Clinical and Experimental Surgery, Saarland University, Homburg, Germany) with an age of 8 months and an average body weight of 25 g were used. For the isolation of NF, inguinal subcutaneous adipose tissue from male GFP^+^ donor mice (C57BL/6-Tg(CAG-EGFP)131Osb/LeySopJ; The Jackson Laboratory, Bar Harbor, ME, USA) with an age of 9 months and a body weight of >30 g was harvested. C57BL/6J mice with a mean age of 5 months and an average body weight of 25 g were equipped with dorsal skinfold chambers. The use of GFP^+^ NF allowed its differentiation from the GFP^−^ host tissue after transplantation.

The animals were kept at a constant room temperature of 22–24 °C and subjected to a 12 h day–night cycle. They had free access to standard pellet chow (Altromin, Lage, Germany) and water. Mice with a dorsal skinfold chamber were housed individually in cages for the entire duration of the experiment.

### 4.2. Anesthesia

All experimental procedures, including the harvesting of adipose tissue, the preparation of the dorsal skinfold chamber as well as stereomicroscopic and intravital fluorescent microscopic imaging, were performed under general anesthesia. This was induced by the intraperitoneal (i.p.) injection of ketamine (100 mg/kg body weight; Ursotamin^®^; Serumwerke Bernburg, Bernburg, Germany) and xylazine (12 mg/kg body weight; Rompun^®^; Bayer, Leverkusen, Germany). For peri-operative pain prevention, the animals received a subcutaneous injection of 10 mg/kg carprofen (Rimadyl^®^; Zoetis Deutschland GmbH, Berlin, Germany) at the beginning of the chamber preparation.

### 4.3. Generation of PRP

For the generation of PRP, 10 C57BL/6J donor mice were anesthetized and a median laparotomy was performed. The inferior vena cava was exposed and the entire volume of blood of each mouse (~1 mL) was extracted via venous puncture. Pooled blood samples were collected in lithium heparin monovettes (Sarstedt, Nümbrecht, Germany). The blood was then centrifuged for 15 min at 110× *g* to separate the PRP from the erythrocyte pellet and platelet-poor plasma. The resulting PRP was stored at −20 °C until further use.

### 4.4. Generation of NF

NF was generated as previously described [19]. Briefly, we excised the subcutaneous adipose tissue from the inguinal region of 12 anesthetized GFP^+^ donor mice, avoiding the inclusion of inguinal lymph nodes. The fat was then washed in 0.9% NaCl solution and cut in standardized samples (dimension: 1 × 1 × 1 mm) by means of a tissue chopper (McIlwain Tissue Chopper, CLE Co., Ltd., Gomshall, UK). These samples were further mechanically shuffled between two syringes using three female-to-female Luer lock connectors with decreasing internal diameters of 2.4, 1.4 and 1.2 mm and 30 passes per connector. At the end of this mechanical emulsification process, the suspension was filtered through a 500 μm pore filter to remove any larger residual tissue aggregates.

### 4.5. Dorsal Skinfold Chamber Model

To guarantee a standardized setting for the investigation of wound healing, full-thickness skin defects were created in dorsal skinfold chambers of 24 C57BL/6 J mice [39]. For the implantation of these chambers, the dorsal skin of the anesthetized animals was carefully shaven and chemically depilated (asid-med depilation cream; Asid Bonz GmbH, Herrenberg, Germany). Subsequently, two titanium frames (Irola Industriekomponenten GmbH & Co. KG, Schonach, Germany) were sutured to the dorsal skin of the animals as described previously in detail [40]. Following a recovery period of 2 days, the mice were anesthetized again and a full-thickness skin defect (diameter: ~4 mm) was prepared in the center of the chamber observation window using a 4 mm dermal biopsy punch (GSK Consumer Healthcare, GMDT, Clocherane, Youghal Road, Dungarvan, Co., Waterford, Ireland) and micro-scissors (Fine Science Tools GmbH, Heidelberg, Germany).

In the PRP + NF group (*n* = 8), each wound was filled with a mixture of 5 µL of PRP and 3 µL of NF. In the PRP control group (*n* = 8), the wounds were filled with a mixture of 5 µL of PRP and 3 µL of phosphate-buffered saline (PBS). The PRP in both groups was then activated with 2 µL of thrombin (10 U/mL dissolved in 10% CaCl_2_; Sigma-Aldrich, Steinheim, Germany). In an additional control group (*n* = 8), the wounds were kept empty and solely filled with 10 µL of PBS. At the end of the procedure, the observation window of each chamber was sealed with a cover glass that was secured to the chamber frame by means of a removable snap ring.

### 4.6. Stereomicroscopy

Wound healing progression was repeatedly analyzed by means of stereomicroscopy on days 0 (day of wound creation), 3, 6, 10 and 14. For this purpose, the anesthetized mice were fixed to a plexiglass stage and the observation window of the chamber was positioned horizontally under the microscope (Leica M651, Wetzlar, Germany). The microscopic images of the wounds were recorded on DVD and analyzed by means of the computer-assisted off-line analysis system CapImage (version 8.5; Zeintl, Heidelberg, Germany). The area of each wound was calculated and expressed as the percentage (%) of the area on day 0.

### 4.7. Intravital Fluorescence Microscopy

Directly after stereomicroscopy, the wounds were additionally imaged by means of intravital fluorescence microscopy to assess morphological and microhemodynamic parameters of tissue perfusion. To enhance contrast, each microscopy was performed after the injection of 0.1 mL of the blood plasma marker, 5% fluorescein-isothiocyanate (FITC)-labeled dextran (150,000 Da; Sigma-Aldrich, Taufkirchen, Germany), into the retrobulbar venous plexus. Imaging was performed using a Zeiss Axiotech fluorescence epi-illumination microscope equipped with 1.25×, 5×, 10× and 20× long-distance objectives (Zeiss, Oberkochen, Germany). The microscopic images were recorded on DVD using a charge-coupled device video camera (FK6990; Pieper, Schwerte, Germany) and analyzed off-line by means of CapImage. The vascularization of the wounds was assessed over time in 6 standardized ROIs in each wound. This analysis included the determination of the number of ROIs containing red blood cell (RBC)-perfused microvessels (% of all analyzed ROIs). Furthermore, we measured the functional microvessel density (cm/cm^2^), i.e., the overall length of all RBC-perfused microvessels per ROI. In addition, the diameter (µm), centerline RBC velocity (µm/s), volumetric blood flow (pL/s) and shear rate (s^−1^) were assessed in up to 5 randomly chosen RBC-perfused microvessels per ROI as described previously in detail [19]. After the last microscopy, the animals were sacrificed via cervical dislocation under anesthesia and tissue specimens were excised and fixed in 4% formaldehyde for further histological and immunohistochemical analyses.

### 4.8. Histology and Immunohistochemistry

Formaldehyde-fixed tissue samples were embedded in paraffin and cut into sections with 3 µm thickness. Sections were stained with hematoxylin and eosin (HE) according to a standard protocol. Additional sections were stained with primary antibodies against perilipin (1:100; Cell Signaling, Danvers, MA, USA), Col I (1:250; Abcam, Cambridge, UK), Col III (1:100; Proteintech, Rosemont, IL, USA), CD31 (1:100; dianova GmbH, Hamburg, Germany), LYVE-1 (1:200; Abcam), GFP (1:100; Rockland, Limerik, PA, USA), CD86 (1:100; Cell Signaling) and CD163 (1:100; Proteintech). As secondary antibodies, we used a horseradish peroxidase-labeled goat anti-rabbit IgG antibody (1:200; dianova GmbH), a goat-anti-rat IgG-Alexa555 antibody (1:100; Molecular Probes, Eugene, OR, USA) and a goat-anti-rabbit IgG-Alexa555 antibody (1:200; Molecular Probes). 3-Amino-9-ethylcarbazole (Abcam) served as chromogen for light microscopic sections. Hoechst 33342 (2 μg/mL: Sigma-Aldrich) was used to stain cell nuclei on immunofluorescence sections.

The sections were analyzed with a BX60 microscope and the imaging software cellSens Dimension (version 1.11, Olympus, Hamburg, Germany). To measure the density (mm^−2^) of CD31^+^ microvessels and LYVE-1^+^ lymph vessels within each wound, their number was counted and divided by the wound area. Furthermore, the fraction (% of all vessels) of GFP^+^ microvessels and lymph vessels was assessed in each wound within the PRP + NF group. The amount of granulation tissue formation and epithelialization (%) were analyzed on sections displaying the largest cross-sectional diameter of the wounds. Granulation tissue formation was assessed as the area of granulation tissue inside the wound / total wound area × 100. Epithelialization was assessed as the length of the newly developed epithelial layer covering the wounds / total wound diameter × 100. In addition, the cellular density (mm^−2^), the total Col I and Col III ratio (Col content in wounds in relation to normal skin), the amount of CD86^+^ and CD163^+^ macrophages (mm^−2^) and the CD163^+^/CD86^+^ ratio were assessed in 2 ROIs in the border zones and 1 ROI in the center of each wound.

### 4.9. Statistical Analysis

We initially tested the data for normal distribution and equal variance. Differences between the three groups were analyzed by one-way ANOVA (parametric data) or ANOVA on ranks (non-parametric data) followed by a Student–Newman–Keuls post hoc test (Prism software version 10.0.3, GraphPad, San Diego, CA, USA). All values were expressed as mean ± standard error of the mean (SEM). Statistical significance was accepted for a value of *p* < 0.05.

## Figures and Tables

**Figure 1 ijms-25-00851-f001:**
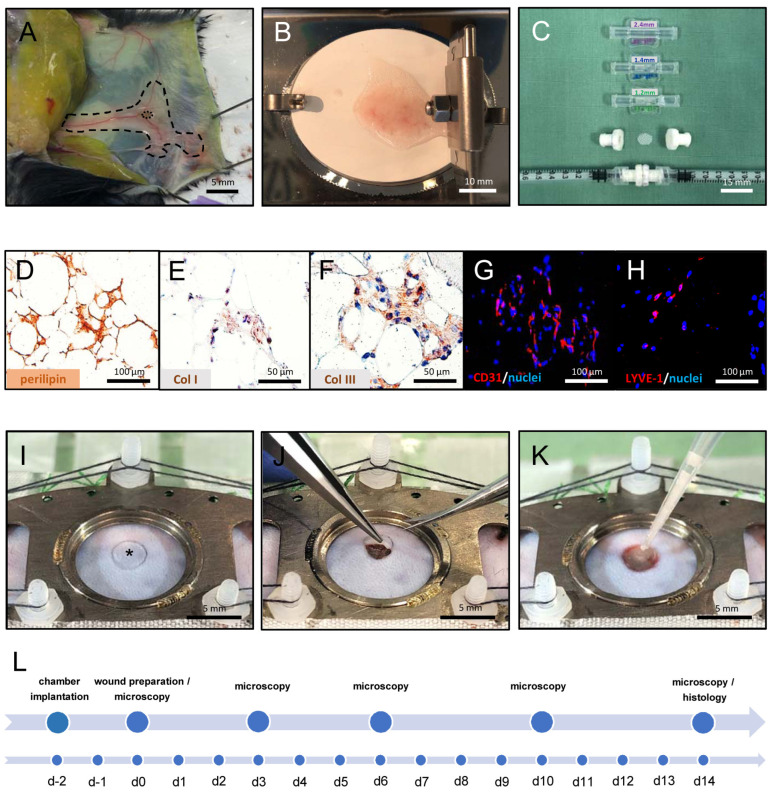
Wound chamber model preparation and immunohistochemical images of nanofat (NF). (**A**) Inguinal subcutaneous adipose tissue of a green fluorescent protein (GFP)^+^ donor mouse (marked by broken line) with an inguinal lymph node (marked by dotted line). (**B**) Harvested inguinal adipose tissue under a tissue chopper. (**C**) From top to bottom: Three connectors (internal diameter: 2.4 mm, 1.4 mm and 1.2 mm) as well as two adaptors with a cell filter (pore size: 500 μm) in between and their combination with two syringes for the mechanical emulsification and filtration of NF. (**D**–**H**) Immunohistochemical characterization of NF, which consists of perilipin^+^ disrupted and intact adipocytes (**D**), collagen (Col) I^+^ (**E**) and Col III^+^ (**F**) extracellular matrix components as well as CD31^+^ microvessels (**G**) and lymphatic vessel endothelial hyaluronan receptor (LYVE)-1^+^ lymph vessels (**H**). (**I**–**K**) Preparation of a full-thickness skin defect (**I**, asterisk) in the observation window of a dorsal skinfold chamber by means of a biopsy punch and a micro-scissors (**I**,**J**). Subsequently, the defect is filled with platelet-rich plasma (PRP) and NF (**K**). (**L**) Experimental protocol of the study. The implantation of the dorsal skinfold chamber is performed on day (d) -2. After 2 days, a full-thickness skin defect is prepared in the center of the chamber and left empty or filled with PRP or PRP + NF. Wound healing is repeatedly analyzed by means of stereomicroscopy and intravital fluorescence microscopy at the indicated time points. On day 14, tissue samples of the wounds are harvested and processed for further histological and immunohistochemical analyses.

**Figure 2 ijms-25-00851-f002:**
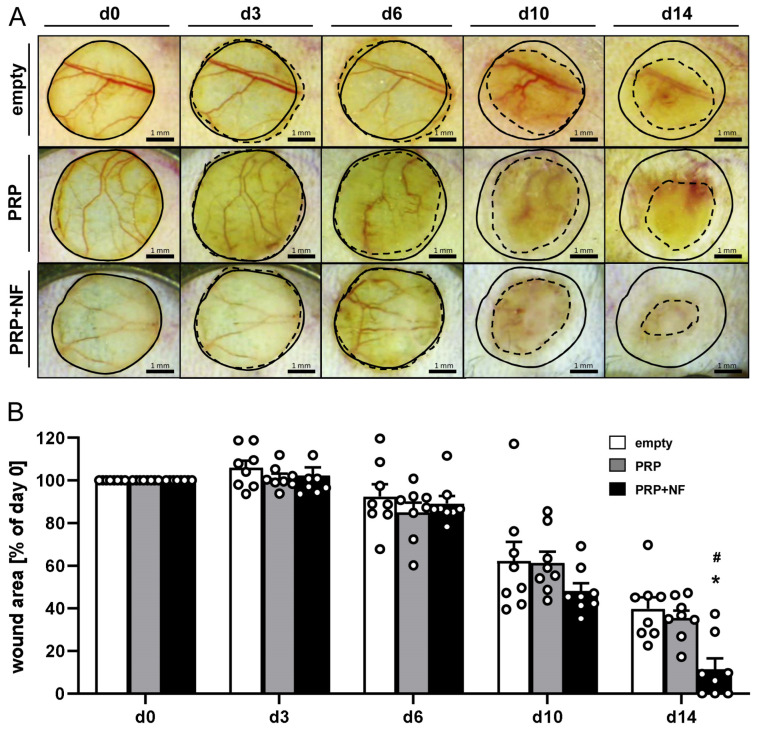
Stereomicroscopic analyses of the wounds. (**A**) Stereomicroscopic images of empty (upper panels), PRP-treated (middle panels) and PRP + NF-treated (lower panels) wounds on days 0, 3, 6, 10 and 14 (initial wound borders = closed lines; wound borders at the indicated time points = broken lines). (**B**) Wound area (% of day 0) of empty (white bars; *n* = 8), PRP-treated (gray bars; *n* = 8) and PRP + NF-treated (black bars; *n* = 8) wounds on days 0, 3, 6, 10 and 14, as assessed by means of stereomicroscopy. Means ± SEM. * *p* < 0.05 vs. empty; # *p* < 0.05 vs. PRP.

**Figure 3 ijms-25-00851-f003:**
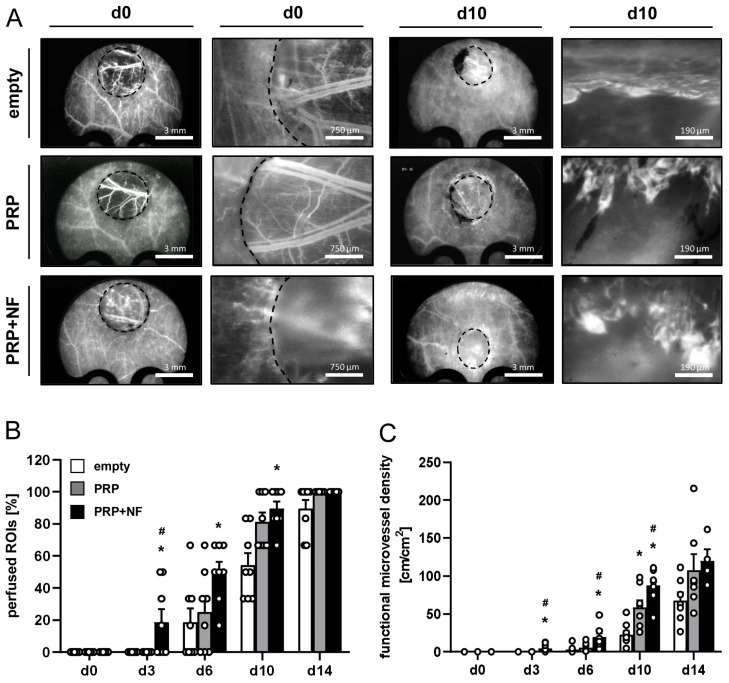
Intravital fluorescence microscopic analyses of the wounds. (**A**) Intravital fluorescence microscopic images of empty, PRP-treated and PRP + NF-treated wounds (left panels: overview; right panels: higher magnification) on days 0 and 10. (**B**,**C**) Perfused regions of interest (ROIs) (**B**, %) and functional microvessel density (**C**, cm/cm²) of empty (white bars, *n* = 8), PRP-treated (gray bars, *n* = 8) and PRP + NF-treated (black bars, *n* = 8) wounds on days 0, 3, 6, 10 and 14, as assessed by means of intravital fluorescence microscopy. Means ± SEM. * *p* < 0.05 vs. empty; # *p* < 0.05 vs. PRP.

**Figure 4 ijms-25-00851-f004:**
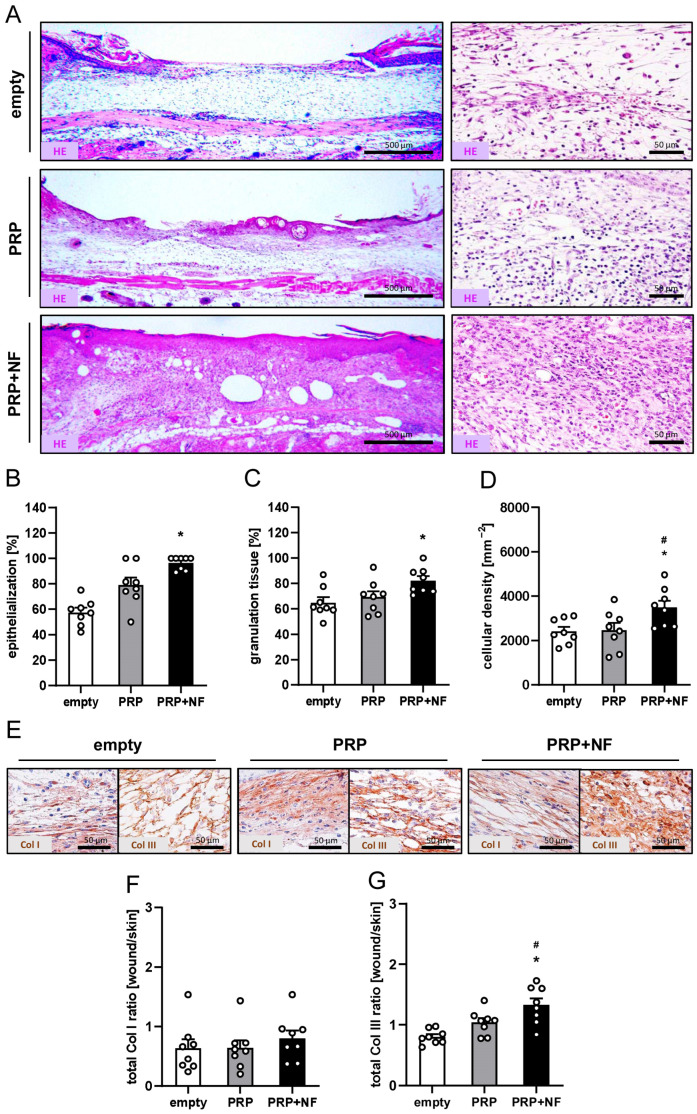
Histological and immunohistochemical analyses of wounds on day 14. (**A**) Hematoxylin–eosin (HE)-stained sections of empty, PRP-treated and PRP + NF-treated wounds (**left panels**: overview; **right panels**: higher magnification) on day 14. (**B**–**D**) Epithelialization (**B**, %), granulation tissue (**C**, %) and cellular density (**D**, mm^−2^) of empty (white bars, *n* = 8), PRP-treated (gray bars, *n* = 8) and PRP + NF-treated (black bars, *n* = 8) wounds on day 14, as assessed by means of histology. Means ± SEM. * *p* < 0.05 vs. empty; # *p* < 0.05 vs. PRP. (**E**) Immunohistochemical sections of empty, PRP-treated and PRP + NF-treated wounds on day 14. The sections were stained with antibodies against Col I and Col III. (**F**,**G**) Total Col I ratio (**F**, wound/skin) and total Col III ratio (**G**, wound/skin) of empty (white bars, *n* = 8), PRP-treated (gray bars, *n* = 8) and PRP + NF-treated (black bars, *n* = 8) wounds on day 14, as assessed by means of immunohistochemistry. Means ± SEM. * *p* < 0.05 vs. empty; # *p* < 0.05 vs. PRP.

**Figure 5 ijms-25-00851-f005:**
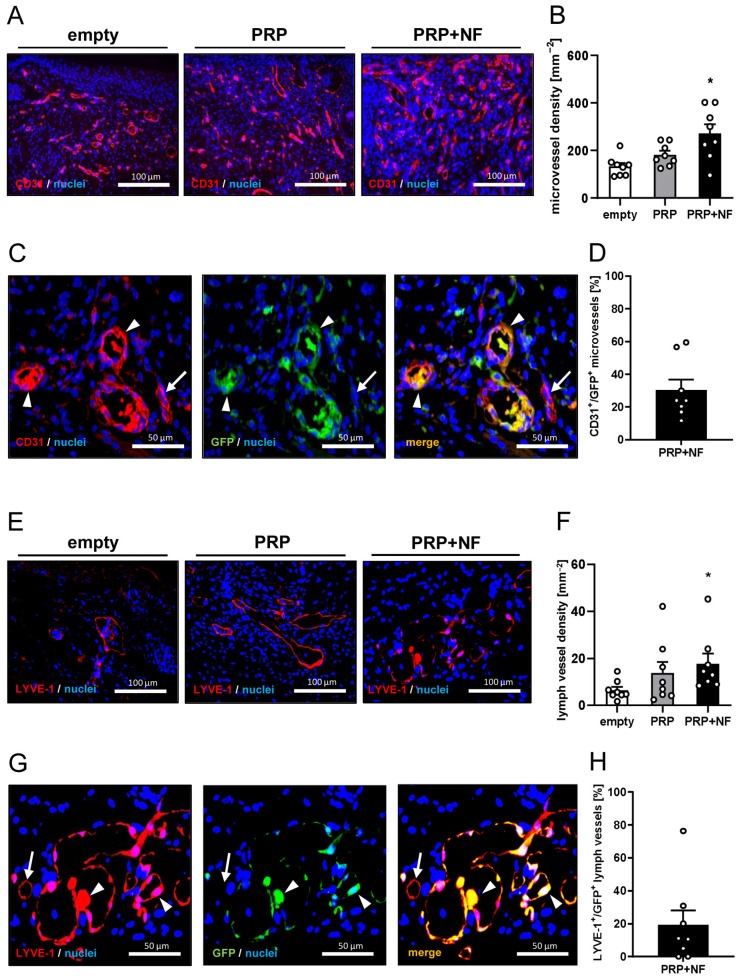
Immunofluorescence analyses of wounds on day 14. (**A**) Immunofluorescence sections of empty, PRP-treated and PRP + NF-treated wounds on day 14. The sections were stained with an antibody against CD31 and Hoechst 33342 to mark cell nuclei. (**B**) Microvessel density (mm^−2^) of empty (white bar, *n* = 8), PRP-treated (gray bar, *n* = 8) and PRP + NF-treated (black bar, *n* = 8) wounds on day 14, as assessed by means of immunofluorescence. Means ± SEM. * *p* < 0.05 vs. empty. (**C**) Immunofluorescence sections of empty, PRP-treated and PRP + NF-treated wounds on day 14. The sections were stained with antibodies against CD31 and GFP to identify CD31^+^/GFP^−^ (arrow) and CD31^+^/GFP^+^ (arrowheads) microvessels as well as Hoechst 33342 to mark cell nuclei. (**D**) CD31^+^/GFP^+^ microvessels (%) within PRP + NF-treated wounds (black bar, *n* = 8) on day 14, as assessed by means of immunofluorescence. Means ± SEM. (**E**) Immunofluorescence sections of empty, PRP-treated and PRP + NF-treated wounds on day 14. The sections were stained with an antibody against LYVE-1 and Hoechst 33342 to mark cell nuclei. (**F**) Lymph vessel density (mm^−2^) of empty (white bar, *n* = 8), PRP-treated (gray bar, *n* = 8) and PRP + NF-treated (black bar, *n* = 8) wounds on day 14, as assessed by means of immunofluorescence. Means ± SEM. * *p* < 0.05 vs. empty. (**G**) Immunofluorescence sections of empty, PRP-treated and PRP + NF-treated wounds on day 14. The sections were stained with antibodies against LYVE-1 and GFP to identify LYVE-1^+^/GFP^−^ (arrow) and LYVE-1^+^/GFP^+^ (arrowheads) lymph vessels as well as Hoechst 33342 to mark cell nuclei. (**H**) LYVE-1^+^/GFP^+^ lymph vessels (%) within PRP + NF-treated wounds (black bar, *n* = 8) on day 14, as assessed by means of immunofluorescence. Means ± SEM.

**Figure 6 ijms-25-00851-f006:**
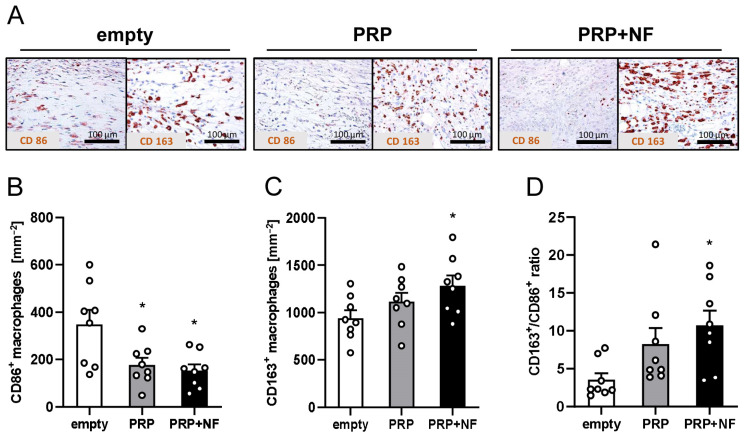
Immunohistochemical analyses of immune cell infiltration into wounds on day 14. (**A**) Immunohistochemical sections of empty, PRP-treated and PRP + NF-treated wounds on day 14. The sections were stained with antibodies against CD86 and CD163 to identify M1- and M2-polarized macrophages. (**B**–**D**) CD86^+^ macrophages (**B**, mm^−2^), CD163^+^ macrophages (**C**, mm^−2^) and CD163^+^/CD86^+^ ratio (**D**) within empty (white bars, *n* = 8), PRP-treated (gray bars, *n* = 8) and PRP + NF-treated (black bars, *n* = 8) wounds on day 14, as assessed by means of immunohistochemistry. Means ± SEM. * *p* < 0.05 vs. empty.

**Table 1 ijms-25-00851-t001:** Diameter (µm), centerline red blood cell (RBC) velocity (µm/s), shear rate (s^−1^) and volumetric blood flow (pL/s) of microvessels within empty, PRP-treated and PRP + NF-treated wounds. Mean ± SEM; * *p* < 0.05 vs. empty; # *p* < 0.05 vs. PRP.

	d0	d3	d6	d10	d14
**Diameter (µm):**					
empty	-	-	24.1 ± 2.2	17.2 ± 0.6	14.3 ± 0.4
PRP	-	-	23.2 ± 1.1	19.6 ± 1.1	15.4 ± 0.8
PRP + NF	-	17.6 ± 0.6	17.9 ± 0.5 *	17.6 ± 0.6	15.6 ± 0.6
**Centerline RBC velocity (µm/s):**					
empty	-	-	201.0 ± 18.1	189.5 ± 20.5	207.7 ± 22.8
PRP	-	-	104.8 ± 17.9 *	213.4 ± 27.7	227.7 ± 34.0
PRP + NF	-	203.6 ± 4.9	226.9 ± 13.2 ^#^	216.5 ± 12.4	223.6 ± 13.6
**Shear rate (s^−1^):**					
empty	-	-	74.5 ± 8.1	92.3 ± 11.4	121.9 ± 15.0
PRP	-	-	51.7 ± 16.3	95.3 ± 13.4	125.4 ± 19.4
PRP + NF	-	101.5 ± 7.7	110.7 ± 9.2 ^#^	101.4 ± 6.8	102.0 ± 9.9
**Volumetric blood flow (pL/s):**					
empty	-	-	58.9 ± 11.8	28.1 ± 3.3	20.7 ± 1.9
PRP	-	-	36.6 ± 7.1	39.3 ± 5.1	27.6 ± 5.3
PRP + NF	-	29.8 ± 3.4	38.9 ± 3.2	35.9 ± 4.1	29.5 ± 6.6

## Data Availability

The data that support the findings of this study are available from the corresponding author upon reasonable request.

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
