# Peer review of "Nanofat Accelerates and Improves the Vascularization, Lymphatic Drainage and Healing of Full-Thickness Murine Skin Wounds"

_ijms, 2024, doi:10.3390/ijms25020851_

Round 1

Reviewer 1 Report

Comments and Suggestions for Authors

The authors prepared nanofat from the adipose tissue of GFP+ donor mice and investigated their ability in healing skin wounds, providing a preliminary illustration of the potential mechanism. Overall, the results were sufficiently presented and discussed in the study. Nevertheless, several doubtful points should be clarified.

Q1. The introduction section should better include additional details about nanofat, including its definition, characteristics, and previous studies highlighting its potential in wound treatment and associated mechanisms.

Q2. Figure 2B, 3B, 3C, 4B-D, 4F, 4G, 6B-D, the discussion should include an analysis of the statistical differences in the investigated parameters between the PRP- and PRP+NF-treated groups.

Q3. Table 1 needs to be revised using a scientifically formatted three-line layout. Moreover, same concerns as Q2 should be acknowledged.

Q4. As demonstrated in figure 3, the functional microvessel density in wound of PRP+NF-treated group was significantly higher than that of the empty group at day 10. Why was the histological and immunohistochemical analysis of wounds not chosen for the same time point, instead of selecting day 14? Besides, to better illustrate the potential mechanism of NF, shouldn't the immunofluorescence analyses of wounds be conducted at day 10?

Q5. Is there any previous literature supporting the current procedure of combining NF with PRP for wound treatment?

Q6. As demonstrated in Lines 212-216, PRP contains various angiogenic growth factors, which likely contribute to wound healing in addition to the potential involvement of NF alone. Therefore, the title of the manuscript should be reconsidered.

Q7. What’s the stability of NF? Is there has any emulsifier involved?

Q8. Concentration of NF and PRP was missing?

Comments:

1. Are there any possibilities to enhance the efficiency of wound healing using NF by optimizing their quantity or PRP concentration (or quantity)?

2. If we attribute the healing mechanism of NF to its significant fraction of viable GFP+ vascular and lymph vessel fragments, what role does fat play in this process?

Author Response

Review of the manuscript ID ijms-2809696 by Limido et al.

Reply to the comments of reviewer 1

We appreciate the fair and constructive comments of the reviewer. In the following, please find our point-by-point reply.

1. Reviewer comment: The introduction section should better include additional details about nanofat, including its definition, characteristics, and previous studies highlighting its potential in wound treatment and associated mechanisms.

Reply: According to the comment of the reviewer, we have included additional details about nanofat in the introduction section of our revised manuscript. The revised paragraph about nanofat now reads as follows:

‘These problems may be overcome by the use of nanofat (NF), which is applied in the field of regenerative and aesthetic medicine for facial rejuvenation, lipomodeling, scar repair, burn wound management and alopecia treatment [Suh et al., 2019; Nepal et al., 2021; Abouzaid et al., 2022; La Padula et al., 2023]. NF is an autologous fat derivative that is generated by mechanical emulsification and filtration to obtain a liquid suspension [Tonnard et al., 2013]. In this suspension most of the mature adipocytes are destroyed [Osinga et al., 2015; Yu et al., 2018]. Accordingly, it is mainly characterized by a high content of extracellular matrix, growth factors, adipose tissue-derived stem cells (ADSC) and microvascular fragments (MVF) [Frueh et al., 2017; Jeyaraman et al., 2021; Weinzierl et al., 2022]. All of these individual components have already been shown to promote tissue formation and the development of new blood-perfused microvascular networks [Schultz and Wysocki, 2009; Sharma et al., 2021; Park et al., 2023]. Therefore, it may be assumed that NF ideally supports the regeneration of skin defects.’

(See lines 54-63; lines 434-439; lines 445-449; lines 457-463; marked in yellow)

References:

Abouzaid, A.M.; El Mokadem, M.E.; Aboubakr, A.K.; Kassem, M.A.; Al Shora, A.K.; Solaiman, A. Effect of autologous fat transfer in acute burn wound management: A randomized controlled study. Burns 2022, 48, 1368-1385.

Frueh, F.S.; Später, T.; Lindenblatt, N.; Calcagni, M.; Giovanoli, P.; Scheuer, C.; Menger, M.D.; Laschke, M.W. Adipose Tissue-Derived Microvascular Fragments Improve Vascularization, Lymphangiogenesis, and Integration of Dermal Skin Substitutes. J Invest Dermatol 2017, 137, 217-227.

Jeyaraman, M.; Muthu, S.; Sharma, S.; Ganta, C.; Ranjan, R.; Jha, S.K. Nanofat: A therapeutic paradigm in regenerative medicine. World J Stem Cells 2021, 13, 1733-1746.

La Padula, S.; Ponzo, M.; Lombardi, M.; Iazzetta, V.; Errico, C.; Polverino, G.; Russo, F.; D'Andrea, L.; Hersant, B.; Meningaud, J.P.; Salzano, G.; Pensato, R. Nanofat in Plastic Reconstructive, Regenerative, and Aesthetic Surgery: A Review of Advancements in Face-Focused Applications. J Clin Med 2023, 12, 4351.

Nepal, S.; Venkataram, A.; Mysore, V. The Role of Adipose Tissue in Hair Regeneration: A Potential Tool for Management? J Cutan Aesthet Surg 2021, 14, 295-304.

Osinga, R.; Menzi, N.R.; Tchang, L.A.H.; Caviezel, D.; Kalbermatten, D.F.; Martin, I.; Schaefer, D.J.; Scherberich, A.; Largo, R.D. Effects of intersyringe processing on adipose tissue and its cellular components: implications in autologous fat grafting. Plast Reconstr Surg 2015, 135, 1618-1628.

Park, G.T.; Lim, J.K.; Choi, E.B.; Lim, M.J.; Yun, B.Y.; Kim, D.K.; Yoon, J.W.; Hong, Y.G.; Chang, J.H.; Bae, S.H.; Ahn, J.Y.; Kim, J.H. Transplantation of adipose tissue-derived microvascular fragments promotes therapy of critical limb ischemia. Biomater Res 2023, 27, 70.

Schultz, G.S.; Wysocki, A. Interactions between extracellular matrix and growth factors in wound healing. Wound Repair Regen 2009, 17, 153-162.

Sharma, P.; Kumar, A.; Dey, A.D.; Behl. T; Chadha, S. Stem cells and growth factors-based delivery approaches for chronic wound repair and regeneration: A promise to heal from within. Life Sci 2021, 268, 118932.

Suh, A.; Pham, A.; Cress, M.J.; Pincelli, T.; TerKonda, S.P.; Bruce, A.J.; Zubair, A.C.; Wolfram, J.; Shapiro, S.A. Adipose-derived cellular and cell-derived regenerative therapies in dermatology and aesthetic rejuvenation. Ageing Res Rev 2019, 54, 100933.

Tonnard, P.; Verpaele, A.; Peeters, G.; Hamdi, M.; Cornelissen, M.; Declercq, H. Nanofat grafting: Basic research and clinical applications. Plast Reconstr Surg 2013, 132, 1017-1026.

Weinzierl, A.; Harder, Y.; Schmauss, D.; Menger, M.D.; Laschke, M.W. Boosting Tissue Vascularization: Nanofat as a Potential Source of Functional Microvessel Segments. Front Bioeng Biotechnol 2022, 10, 820835.

Yu, Q.; Cai, Y.; Huang, H.; Wang, Z.; Xu, P.; Wang, X; Zhang, L.; Zhang, W.; Li, W. Co-Transplantation of Nanofat Enhances Neovascularization and Fat Graft Survival in Nude Mice. Aesthet Surg J 2018, 38, 667-675.

2. Reviewer comment: Figure 2B, 3B, 3C, 4B-D, 4F, 4G, 6B-D, the discussion should include an analysis of the statistical differences in the investigated parameters between the PRP- and PRP+NF-treated groups.

Reply: According to the comment of the reviewer, we have included additional information about the differences in the investigated parameters between the PRP- and PRP+NF-treated groups in the discussion section of the revised version of our manuscript, which reads as follows:

‘In fact, in the PRP+NF group we detected a significantly higher number of perfused ROIs on day 3 as well as an increased functional microvessel density between day 3-10 and centerline RBC velocity and shear rate on day 6 when compared to the PRP group.’

(See lines 223-226; marked in yellow)

‘We detected significantly higher amounts of Col III in PRP+NF-treated wounds on day 14 when compared to PRP-treated and empty wounds.’

(See lines 252-253; marked in yellow)

3. Reviewer comment: Table 1 needs to be revised using a scientifically formatted three-line layout. Moreover, same concerns as Q2 should be acknowledged.

Reply: According to the comment of the reviewer, we have separated the individual parameters within revised Table 1 (marked in yellow) by additional lines to indicate that each parameter is shown in the requested scientifically formatted three-line layout. Moreover, we have addressed the differences between PRP+NF-treated wounds and PRP-treated wounds in the discussion section of our revised manuscript version (see lines 223-226; marked in yellow).

4. Reviewer comment: As demonstrated in figure 3, the functional microvessel density in wound of PRP+NF-treated group was significantly higher than that of the empty group at day 10. Why was the histological and immunohistochemical analysis of wounds not chosen for the same time point, instead of selecting day 14? Besides, to better illustrate the potential mechanism of NF, shouldn't the immunofluorescence analyses of wounds be conducted at day 10?

Reply: We agree with the reviewer that an additional histological and immunohistochemical analysis on day 10 would have been interesting. However, at this earlier time point, we did not yet detect a significantly reduced wound area of PRP+NF-treated wounds when compared to PRP-treated and empty wounds (see Fig. 2B), which was the main parameter of our study. Therefore, all histological and immunohistochemical analyses were performed on day 14. 

5. Reviewer comment: Is there any previous literature supporting the current procedure of combining NF with PRP for wound treatment?

Reply: There is indeed no previous literature supporting the current procedure of combining NF and PRP for wound treatment. Because we feel that this is a highly promising novel approach, we emphasize now this aspect in the final conclusion paragraph of our discussion section in the revised version of our manuscript (see line 269; marked in yellow).

6. Reviewer comment: As demonstrated in Lines 212-216, PRP contains various angiogenic growth factors, which likely contribute to wound healing in addition to the potential involvement of NF alone. Therefore, the title of the manuscript should be reconsidered.

Reply: Because for most of the analyzed parameters we did not detect any statistically significant differences between PRP-treated and empty wounds, we feel that the positive effects on wound healing were mainly caused by NF and not by the angiogenic growth factors within PRP. Therefore, we have decided to keep the original title of our manuscript. 

7. Reviewer comment: What’s the stability of NF? Is there has any emulsifier involved?

Reply: As explained in the revised introduction of our manuscript (see lines 56-58; marked in yellow), NF is simply generated by mechanical emulsification and filtration, which results in a liquid suspension without the use of any external emulsifier.

8. Reviewer comment: Concentration of NF and PRP was missing?

Reply: Because PRP as well as NF are both highly heterogenous consisting of different components, it is not possible to calculate an exact concentration of these biological products. Therefore, we have decided to provide the volumes of these compounds that were administered to the wounds in the materials and methods part of our manuscript. This paragraph is marked in yellow in the revised version of our manuscript (see lines 327-332).

Additional comments:

1. Reviewer comment: Are there any possibilities to enhance the efficiency of wound healing using NF by optimizing their quantity or PRP concentration (or quantity)?

Reply: There may be the possibility to further optimize the wound healing process by PRP+NF. However, for our experiments we already used an ideal ratio between PRP and NF, which still guaranteed a gel-like consistency of the mixture and, thus, an adequate stability of NF inside the wounds. Higher amounts of NF may impair this balance, resulting in a non-stable wound dressing.  

2. Reviewer comment: If we attribute the healing mechanism of NF to its significant fraction of viable GFP+ vascular and lymph vessel fragments, what role does fat play in this process?

Reply: As explained in the revised introduction of our manuscript, NF is a suspension, in which most of the mature adipocytes are destroyed. However, besides vascular and lymph vessel fragments, it still contains fat-derived growth factors, stem cells and extracelullar matrix components, which all may contribute to an improved wound healing process. 

Reviewer 2 Report

Comments and Suggestions for Authors

This MS follows on from previous work by the same group (ref 13) where the preparation and utility of nanofat for tissue repair was demonstrated. The present work includes the combined use of PRP and nanofat to show faster healing of full thickness wounds in mice. Donor tissue was from GFP+ mice, enabling identification of donor cells in the recipient. Improved healing was attributed to the latter, as well as conversion of macrophages from M1 to M2 phenotype.

The procedure was clearly described. Fig 2 showed a greatly reduced wound closure time for the combination. Unfortunately there was no data for NF alone, particularly in view of ref 20. Had NF alone been shown to be ineffective in the authors’ previous publication? The histological and immunological analyses also did not show data for NF alone.  The authors mention a technical issue with the larger wound size, but couldn’t cell-free serum or a gel have been used to obtain the desired consistency?

The analyses were relevant and conducted well. The results support the use of adipose-derived stem cells to improve wound healing.

Author Response

Review of the manuscript ID ijms-2809696 by Limido et al.

Reply to the comments of reviewer 2

We appreciate the fair and constructive comments of the reviewer. In the following, please find our point-by-point reply.

1. Reviewer comment: The procedure was clearly described. Fig 2 showed a greatly reduced wound closure time for the combination. Unfortunately, there was no data for NF alone, particularly in view of ref 20. Had NF alone been shown to be ineffective in the authors’ previous publication? The histological and immunological analyses also did not show data for NF alone. The authors mention a technical issue with the larger wound size, but couldn’t cell-free serum or a gel have been used to obtain the desired consistency?

Reply: As mentioned in our manuscript, treatment of larger wound beds with NF alone would not be possible, because this biological product is a liquid suspension that is not stable. Accordingly, a gel is necessary for its proper fixation. We agree with the reviewer that we could have also used another cell-free gel for our proof-of-principle experiments. However, we explicitly decided to use PRP, because it is already well-established in clinical practice as an autologous biological product that can be rapidly harvested by minimal-invasive procedures from patients. Therefore, the translation of our approach into the clinical setting may be much easier than approaches involving artificial non-autologous gels.